# Modified Nimo Nanoparticles for Efficient Catalytic Hydrogen Generation from Hydrous Hydrazine

**Ying Liu** [1,2,3,4]**, Huan Zhang** [4]**, Cong Ma** [4,]***and Nan Sun** [5,]*

1   School of Chemical Engineering and Technology, Tianjin University, Tianjin 300072, China
2   National Engineering and Research Centre of Distillation Technology, Tianjin University,
    Tianjin 300072, China
3   Peiyang National Distillation Technology Engineering Company Limited, Tianjin 300072, China
4   State Key Laboratory of Separation Membranes and Membrane Processes, School of Environmental Science
    and Engineering, Tianjin Polytechnic University, Tianjin 300387, China
5   School of Water Conservancy and Civil Engineering, Northeast Agricultural University,
    Harbin 150030, China
*   Correspondence: macong_0805@126.com (C.M.); sunnan@neau.edu.cn (N.S.)

**Abstract:** Precious metal-free NiMoM (M = $Pr_2O_3$, $Cu_2O$) catalysts have been synthesized through a simple coreduction method, without any surfactant or support material, and characterized using X-ray diffraction (XRD), transmission electron microscopy (TEM), and X-ray photoelectron spectroscopy (XPS). The resultant $Pr_2O_3$- or $Cu_2O$-modified NiMo catalysts exhibit different structures, which is due to a difference in the synergistic effects of NiMo and the modifying elements. $NiMoPr_2O_3$ has an amorphous structure, with low crystallinity and uniform particle dispersion, while $NiMo@Cu_2O$ adopts the core–shell structure, where the core and shell are synergistic with each other to promote electron transfer efficiency. The support material-free nanocatalysts $Ni_9Mo_1(Pr_2O_3)_{0.375}$ and $Ni_4Mo@Cu_2O$ are both highly efficient compared with bimetallic NiMo catalysts, in terms of hydrogen generation from hydrous hydrazine ($N_2H_4·H_2O$) at 343 K, with total turnover frequencies (TOFs) of 62 $h^{-1}$ and 71.4 $h^{-1}$, respectively. Their corresponding activation energies (Ea) were determined to be 43.24 kJ $mol^{-1}$ and 46.47 kJ $mol^{-1}$, respectively. This is the first report on the use of Pr-modified NiMo and core–shell $NiMo@Cu_2O$ catalysts, and these results may be used to promote the effective application of noble metal-free nanocatalysts for hydrogen production from hydrous hydrazine.

**Keywords:** NiMoM (M = $Pr_2O_3$, $Cu_2O$) catalysts; hydrous hydrazine; hydrogen production

## 1. Introduction

It is expected that energy consumption and associated environmental issues will continue to grow dramatically in the near future. As an environmentally friendly fuel, hydrogen has the advantages of high combustion heat and high combustion speed. In addition to generating water and a little hydrogen azide during the combustion process, it will not produce substances that are harmful to the environment such as carbon oxides and hydrocarbons compounds. Hydrogen has been proposed as crucial to ensuring secure and sustainable energy development [1–3]. However, the search for effective hydrogen storage materials and methods for hydrogen generation remains a challenge.

In recent years, chemical hydrogen storage has become a popular approach to overcoming the barriers of hydrogen delivery. Hydrous hydrazine ($N_2H_4·H_2O$), which is a liquid at room temperature, is a promising hydrogen carrier due to its relatively low cost, high hydrogen density (8.0 wt. %), low molecular weight (50.06 g $mol^{-1}$) [4,5], and the advantages of only producing $H_2$ and $N_2$ in its decomposition reactions (pathway 1: $N_2H_4 \rightarrow N_2 + 2H_2$). However, to maximize the usability of

$N_2H_4 \cdot H_2O$ in hydrogen storage, we should avoid the undesirable generation of ammonia (pathway 2: $3N_2H_4 \rightarrow N_2 + 4NH_3$). Although ammonia can be burned, not only generates less heat than that of $H_2$ as fuel, but also produces harmful gases like NO. Catalysis plays a critical role in the design of efficient processes and systems able to exploit the advantages of the starting materials, while minimizing waste generation and energy requirements [6]. Several recent studies found that Ni-based catalysts promoted by noble metals, such as Ni-Rh [7–9], Ni-Pt [10–13], Ni-Pd [14–16], and Ni-Ir exhibit superior catalytic performances in the decomposition of hydrous hydrazine, with more than 90% $H_2$ selectivity [17,18]. In fact, although nanocatalysts formed from noble metals and nickel exhibit good catalytic performance, high associated costs limit their use. Other studies have therefore employed nonprecious metals instead of precious metals. He et al. [19] modified Ni nanoparticles with a small amount of $CeO_2$ (8.0 mol %), resulting in significant enhancement of the turnover frequency (TOF) and $H_2$ selectivity. Men et al. [20] synthesized catalysts using $CeO_x$-modified NiFe deposited on reduced graphene oxide (rGO), which exhibited good catalytic characteristics. Manukyan et al. [21] selected copper nanoparticles as the support material, and synthesized a NiFe/Cu catalyst. Copper may be an efficient support material for nickel in the water–gas shift reaction, as it shows high activity and stability. Wang et al. [22] successfully synthesized a $Cu@Fe_5Ni_5$ catalyst, which exhibited high activity in terms of complete $N_2H_4 \cdot H_2O$ decomposition. This is due to the electronic coupling between the core and shell metals. Noble metal-free modified Ni-based catalysts not only facilitate the use of $N_2H_4$ as a chemical hydrogen-producing material, but also provide a potential development path for more rare earth-doped catalysts.

Yang et al. [23] reported that a NiMo catalyst catalyzed the hydrogen production of ammonium borane, and that the electronic interactions between the NiMo enable the catalyst to achieve complete decomposition of the ammonia borane within 2 min. We report a simple coreduction method for preparing NiMo catalysts, which are modified with rare earth elements and nonprecious metals under room temperature, without any support material or surfactant. Pr and Ce are the same as lanthanides elements, with similar chemical structures and properties, although most studies employ Ce as the raw material. In order to further evaluate the value of rare earth elements and their wide application, we employed a Pr-modified NiMo catalyst. As a transition metal, Cu is often the main active element in catalysis. A Cu-modified NiMo catalyst is used to synergize the elements to further improve catalytic performance, and the catalytic properties of the $Ni_{1-x}Mo_x(Pr_2O_3)_y$ and $Ni_{1-x}Mo_x@(Cu_2O)_y$ materials are investigated and compared. The effects of reaction temperature, concentrations of NaOH, and the kinetics of $N_2H_4 \cdot H_2O$ decomposition are also evaluated in this work.

## 2. Results and Discussion

### 2.1. Physical Characteristics of $Ni_{1-x}Mo_x(Pr_2O_3)_y$ and $Ni_{1-x}Mo_x@(Cu_2O)_y$ Nanocatalysts

The one-pot coreduction process provides simple, effective, and readily scalable preparation of nanocatalysts. In the present study, we employ this method to prepare a series of $Ni_{1-x}Mo_x(Pr_2O_3)_y$ and $Ni_{1-x}Mo_x@(Cu_2O)_y$ catalysts, where $x$ represents the molar portion of Mo and $y$ represents the molar portion of the third material ($Cu_2O$ or $Pr_2O_3$). It should be noted that in the preparation of catalysts, the total masses of Ni and Mo are controlled, to a certain extent. After the modification of the NiMo catalysts, $Ni_9Mo_1 (Pr_2O_3)_y$, and $Ni_8Mo_2@(Cu_2O)_y$ refer to the $Ni_{0.9}Mo_{0.1} (Pr_2O_3)_y$ and $Ni_{0.8}Mo_{0.2}@(Cu_2O)_y$ nanocatalysts.

X-ray diffraction (XRD) is used to analyze the crystalline structures of the prepared catalysts. Figure 1a presents the XRD patterns of $Ni_9Mo_1(Pr_2O_3)_y$ and the bimetallic $Ni_{0.2}Pr_{0.015}$, $Mo_{0.2}Pr_{0.015}$, and $Ni_9Mo_1$ catalysts. The diffraction peak of Ni (PDF#04-0850) was detected in all the catalyst samples, with the exception of the $Mo_{0.2}Pr_{0.015}$ catalyst, at $2\theta = 44.5°$, corresponding to the Ni(111) crystal plane peak. The face-centered cubic (FCC) configuration of nickel has three crystal faces—(111), (200), and (220)—and here the diffraction peak only corresponds to the (111) crystal plane, indicating that the prepared catalysts have low crystallinity and are a mixture of crystalline and amorphous structures.

Compared with the $Ni_9Mo_1$ catalyst, $Ni_9Mo_1(Pr_2O_3)_y$ showed much weaker Ni peaks, with the (111) peak intensities of the $Ni_9Mo_1(Pr_2O_3)_y$ samples decreasing gradually with increasing $Pr_2O_3$ content. This suggests that the incorporation of Pr into the catalyst may perturb the crystal growth of the Ni nanocrystallite. When the doping ratio of Pr reaches 0.375, the diffraction peaks of Ni are no longer weakened, which indicates that the catalysts of $Ni_9Mo_1(Pr_2O_3)_y$ can reach a point where there is a coexistence of crystalline and amorphous states, under optimal conditions [21,22]. However, as shown in Figure 1b, the XRD pattern of $Ni_8Mo_2@(Cu_2O)_y$ contains prominent diffraction peaks at $2\theta = 36.4°$, 41.3°, 61.3°, and 73.5°, which can be indexed to diffractions of the (111), (200), (220), and (311) planes of $Cu_2O$ (PDF#05-0667), respectively. The diffraction peaks of $Cu_2O$ strengthened with increasing Cu content. According to the transmission electron microscope (TEM) data (Figure 2f), $Cu_2O$ forms a crystalline state on the outer layer of the catalyst, while internal NiMo crystallinity is low, and so no diffraction peak is observed for Ni. These results clearly indicate that the $Ni_8Mo_2@(Cu_2O)_y$ alloy nanocatalyst is not a simple physical mixture of individual metal nanoparticles, but a coupling between the shell and the nucleus [24].

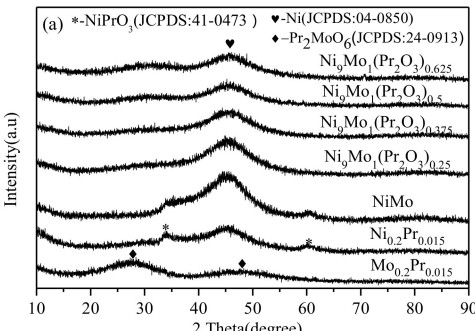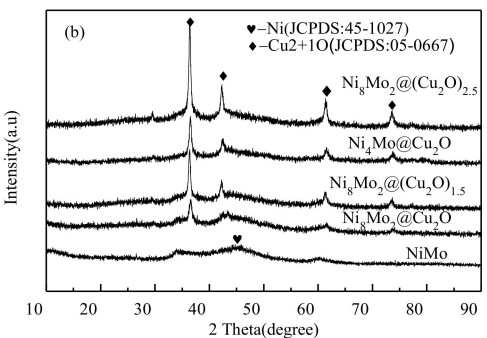

**Figure 1.** (**a**) XRD patterns of $Ni_9Mo_1(Pr_2O_3)_y$ (y = 0, 0.25, 0.375, 0.5, 0.6125) and $Ni_{0.2}Pr_{0.15}$ and $Mo_{0.2}Pr_{0.15}$ catalyst samples. (**b**) XRD patterns of $Ni_4Mo@(Cu_2O)_y$ (y = 0, 1, 1.5, 2, 2.5) catalyst samples.

The TEM images of NiMo, $Ni_9Mo_1(Pr_2O_3)_{0.375}$, and $Ni_4Mo@Cu_2O$, together with the corresponding particle size distributions, are presented in Figure 2. It can be seen from Figure 2a,g that the NiMo catalyst is irregularly dispersed, with a particle size range between 6.5 and 6.9 nm, which is significantly higher than those of the $Ni_9Mo_1(Pr_2O_3)_{0.375}$ and $Ni_4Mo@Cu_2O$ catalysts (5.5–5.9 nm and 6.0–6.3 nm, respectively). The regions with lattice distribution and amorphous states can be clearly seen in high-resolution TEM (HRTEM) images (Figure 2b). According to fast Fourier-transform (FFT) analysis, the lattice spacing of 0.203 nm is consistent with that of the Ni(111) crystal surface, indicating that the crystalline domain is Ni and the amorphous part is formed through NiMo codoping. The $Pr_2O_3$-doped catalyst still has an irregular granular structure after doping, as shown in Figure 2c. According to HRTEM (Figure 2d) observations and FFT analysis, the $Ni_9Mo_1(Pr_2O_3)_{0.375}$ nanocatalyst has a crystalline structure with a lattice fringe distance of 0.208 nm, which is slightly larger than that of the (111) plane of FCC Ni (0.203 nm), indicating that the presence of the atom of Pr in the crystal lattice of Ni increases the lattice spacing. This is consistent with XRD analysis showing the weakening of the diffraction peak after the addition of Pr. At the same time, the Ni lattice expands and distorts, creating more oxygen defects that trap more electrons at the bottom of the conduction band. It can be seen from the Figure 2e that the Cu-modified NiMo catalyst has a transparent coating at the edge and an amorphous structure inside. The corresponding HRTEM image (Figure 2f), suggests that the lattice spacing of the surrounding transparent particles is 0.19 nm, consistent with that of the crystal surface of $Cu_2O(111)$, which indicates that the shell formed in the catalyst is comprised of $Cu_2O$. No lattice fringes were observed at the core, indicating that the core is amorphous, which is consistent with the XRD data showing only the diffraction peak of $Cu_2O$. This implies that a core–shell structure is formed, with NiMo as the nucleus and $Cu_2O$ as the shell. Based on the above analyses, the various modified NiMo nanocatalysts form different structures, which is due to the different configurations of extranuclear electrons in the Pr and

Cu elements. The 5d empty orbit of Pr is a candidate for electron transfer in the synthesized catalyst, which disturbs the Ni electron configuration and causes the crystal shape to change [20,21]. While Ni, Cu, and Mo are transition metals, their different oxidation abilities play an important role in the formation of the structure [25].

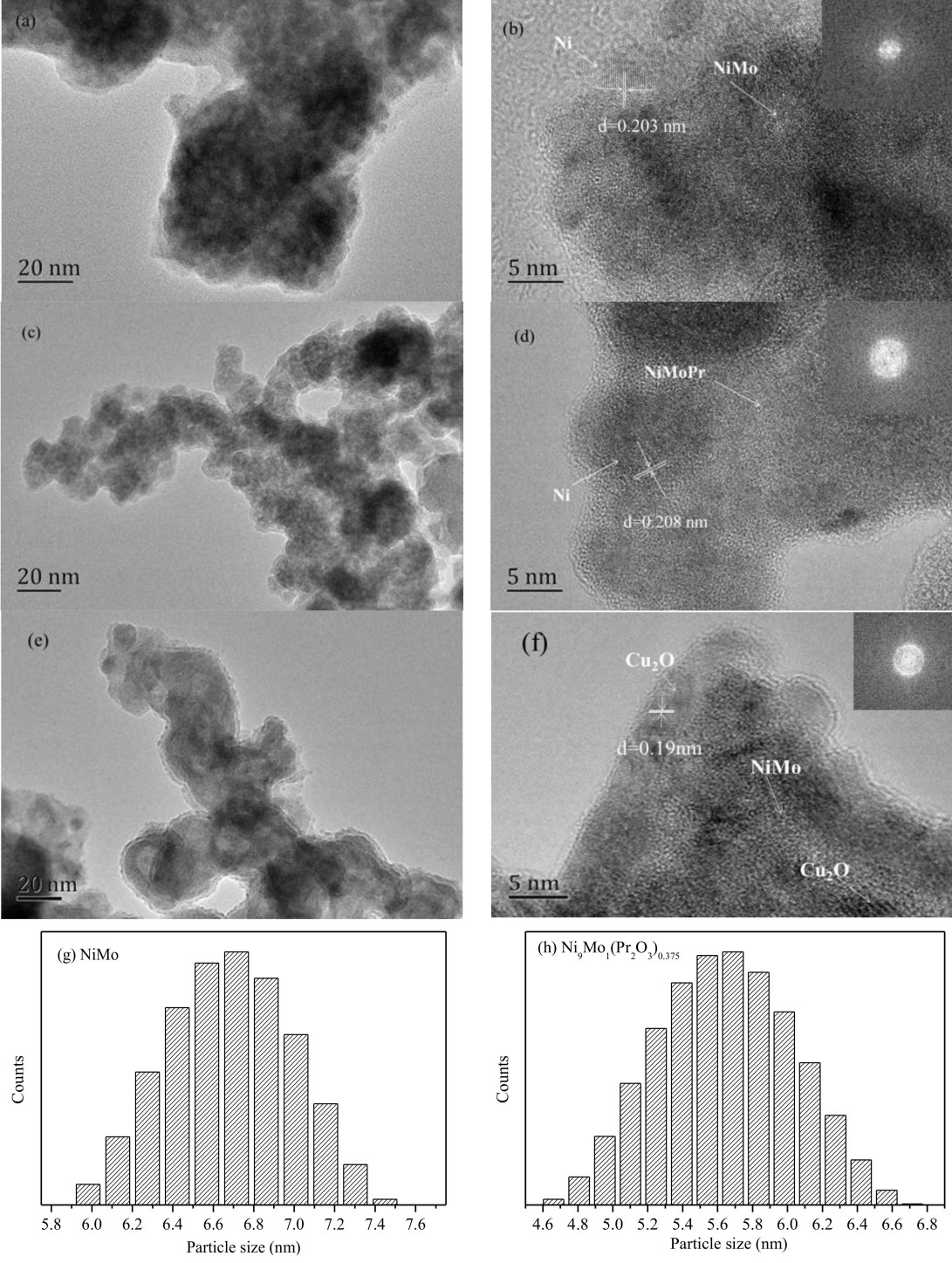

**Figure 2.** *Cont.*

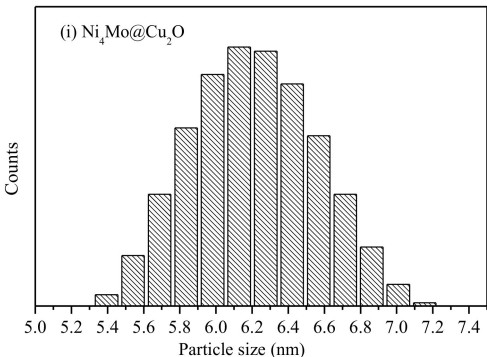

**Figure 2.** (**a**) TEM images of $Ni_9Mo_1$ catalyst, (**b**) a HRTEM image of $Ni_9Mo_1$ catalyst, (**c**) TEM images of $Ni_9Mo_1(Pr_2O_3)_{0.375}$ catalyst, (**d**) a HRTEM image of $Ni_9Mo_1(Pr_2O_3)_{0.375}$ catalyst, (**e**) TEM images of $Ni_4Mo@Cu_2O$ catalyst, (**f**) a HRTEM image of $Ni_4Mo@Cu_2O$ catalyst, (**g**) metal particles size distribution of $Ni_9Mo_1$, (**h**) metal particles size distribution of $Ni_9Mo_1(Pr_2O_3)_{0.375}$, and (**i**) metal particles size distribution of $Ni_4Mo@ Cu_2O$.

Employing a coreduction strategy to form $Ni_9Mo_1(Pr_2O_3)_{0.375}$ and $Ni_4Mo@Cu_2O$ catalysts yields high-performance structures, which exhibited superior and stable catalytic activity in terms of $N_2H_4·H_2O$ selective decomposition to generate $H_2$ under ambient conditions. However, the mechanistic reason for the catalytic performance enhancement that arises as a result of adding Pr and Cu remains unclear. To gain insight into this mechanism, we conducted X-ray photoelectron spectroscopy (XPS) analysis of the NiMo, $Ni_9Mo_1(Pr_2O_3)_{0.375}$, and $Ni_4Mo@Cu_2O$ catalysts. The XPS spectra of NiMo, $Ni_9Mo_1(Pr_2O_3)_{0.375}$, and $Ni_4Mo@Cu_2O$ in the Ni 2p, Mo 3d, Pr 3d, and Cu 2p regions, respectively, are shown in Figure 3. From the XPS spectra, it is apparent that Ni (Figure 3a) is present in the metallic and oxidation state as NiO and $Ni(OH)_2$ [20,26]. The main Ni peaks for $Ni_9Mo_1(Pr_2O_3)_{0.375}$ and $Ni_4Mo@Cu_2O$ are positively shifted to higher values, compared to that of undoped NiMo. This suggests that Ni plays an important role in the electron transfer process. Doping with Cu leads to a positive shift in the binding energies of the metallic Ni (Ni 2p3/2 852.6 eV), which may be due to a decrease in the electron density and an increase in metal center d-band vacancies [27,28]. However, after Pr doping, the binding energy of Ni 2p3/2 was slightly shifted from 855.3 to 855.8 eV, which indicates that Pr doping of the NiMo catalyst has a less significant effect on Ni binding energy than in the case of $Ni_4Mo@Cu_2O$. The Mo species in NiMo (Figure 3b), $Ni_9Mo_1(Pr_2O_3)_{0.375}$ (Figure 3c), and $Ni_4Mo@Cu_2O$ (Figure 3d) are found to exist primarily in the form of oxidation states. Compared with the 3d5/2 and 3d3/2 peaks that appear at 232.1 and 235.15 eV in NiMo, which are attributed to Mo (VI) species such as $MoO_3$ and MoOx, the corresponding peaks for $Ni_9Mo_1(Pr_2O_3)_{0.375}$ and $Ni_4Mo@Cu_2O$ are shifted to higher binding energies, indicating that Mo can act as an electron donor for atoms of Ni and Cu. From Figure 3e, the prominent peak at 932.9 eV is assigned to Pr (III) 3d5/2, indicating that $NaBH_4$ did not reduce $Pr^{3+}$ to Pr in $Ni_9Mo_1(Pr_2O_3)_{0.375}$ during catalyst preparation, and the three elements and their oxides are doped with each other to form irregular particles [29,30]. According to the electrode potential, $Ni^{2+}$ can be readily reduced to Ni by $NaBH_4$, but it is relatively difficult to reduce $Mo^{6+}$ to Mo, due to the lower reduction potentials of $Mo^{6+}/Mo$. After the addition of $Ni^{2+}$ and $Mo^{6+}$, $Ni^{2+}$ is first reduced by $NaBH_4$ due to its higher potential. However, the oxidizing ability of $Cu^{2+}$ is stronger than that of $Ni^{2+}$, and $Cu^{2+}$ was reduced first during the reduction, shown in the XPS results in Figure 3f as $Cu_2O$ (Cu 2p 3/2 932.9 eV, Cu 2p 1/2 953.2 eV) and CuO (Cu 2p 3/2 935.4 eV), together with the satellite peaks, while Ni 2p is only shown as Ni (II) [31].

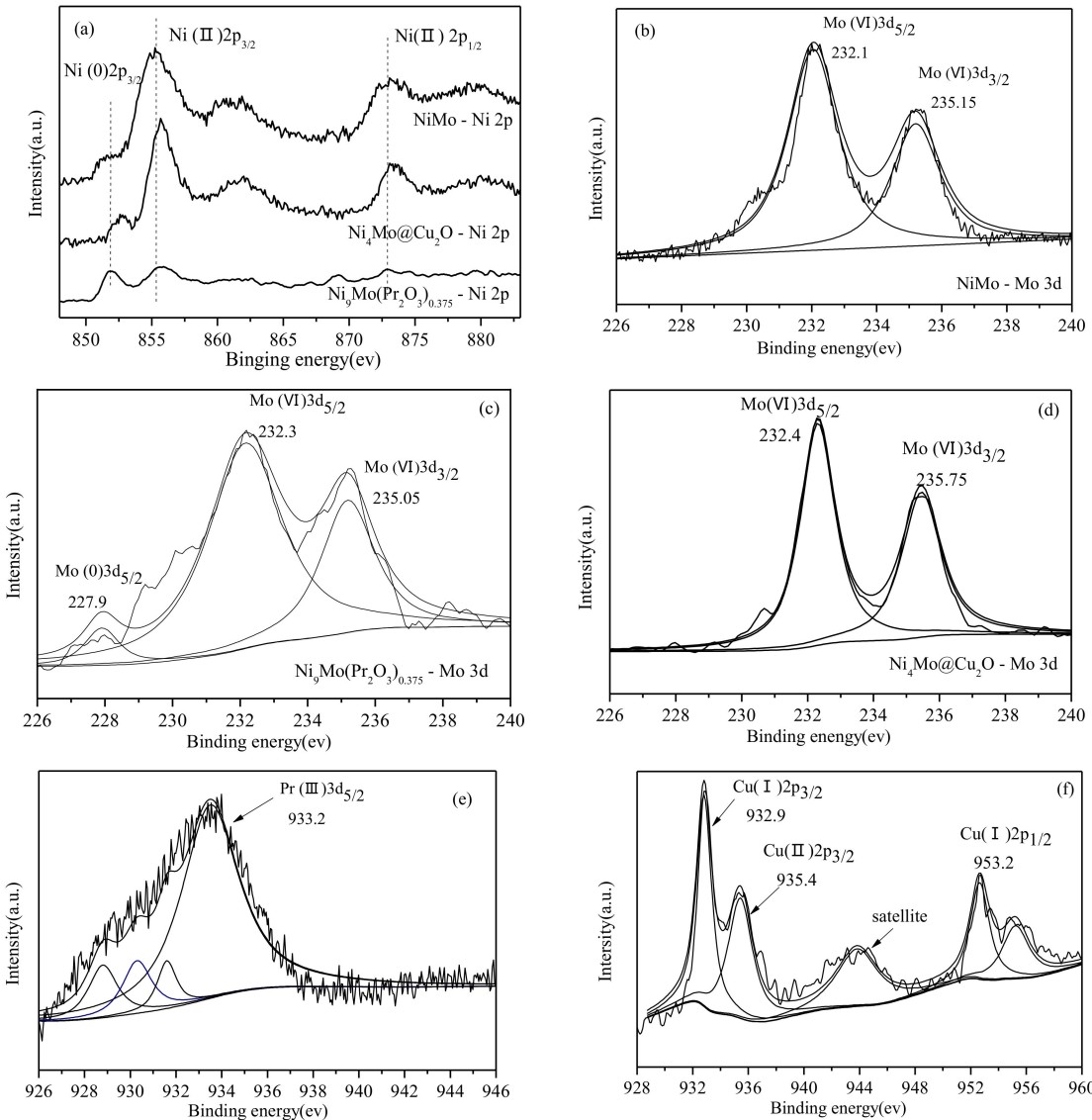

**Figure 3.** X-ray photoelectron spectra of the as-synthesized $Ni_9Mo_1$, $Ni_9Mo_1(Pr_2O_3)_{0.375}$, and $Ni_4Mo@Cu_2O$: (**a**) Ni 2p, (**b**), (**c**), (**d**) Mo 3d, (**e**) Pr 3d, and (**f**) Cu 2p.

## 2.2. Catalytic Activities of $Ni_{1-x}Mo_x(Pr_2O_3)_y$ and $Ni_{1-x}Mo_x@(Cu_2O)_y$ Nanocatalysts

The catalytic performances of the as-synthesized nanocatalysts, in terms of hydrogen generation from $N_2H_4 \cdot H_2O$, are shown in Figure 4. Reactions were initiated by introducing $N_2H_4 \cdot H_2O$ into the reaction flask with vigorous shaking, at a specified temperature. The catalytic performances of the catalysts were evaluated in a typical water-filled gas burette system. As show in Figure 4a, elemental nickel and elemental molybdenum have no catalytic activity, and with the increase of NiMo ratio, the catalytic activity appears to initially increase, and then decrease. A NiMo ratio of 9:1 produces the best catalytic effect, with $n(N_2+H_2)/n(N_2H_4)$ at 1.9, but the requirement for the highly efficient catalytic hydrogen production of hydrazine hydrate is not achieved. It can be seen from Figure 4b,c that even adding a small amount of Pr or Cu significantly improved the performances of catalysts. When the molar ratios of Ni:Mo:Pr and Ni:Mo:Cu were 9:1:0.75 and 8:2:2, respectively (the total amounts of Ni and Mo were 2 mL, 0.1mol $L^{-1}$, respectively), the synthesis of $Ni_9Mo_1(Pr_2O_3)$ and $Ni_4Mo@Cu_2O$ resulted in high hydrogen production efficiency. As shown in Figure 4b, the value of a stoichiometric amount of $H_2$ and $N_2$ was 2.8, and the TOF was 62 $h^{-1}$, for the decomposition of $N_2H_4 \cdot H_2O$ by $Ni_9Mo_1(Pr_2O_3)_{0.375}$. Meanwhile, the $Ni_4Mo@Cu_2O$ with a core–shell structure produced

a stochiometric amount of $H_2$ equal to 2.91 under the same conditions, with a TOF of 71.4 $h^{-1}$, which is higher than the majority of reported values shown in Table 1. The excellent catalytic performance of NiMo catalysts doped with $Pr_2O_3$ or $Cu_2O$ should be primarily attributed to a modification of the electronic structure in the as-synthesized catalysts through interactions between the NiMo and M (M = $Pr_2O_3$, $Cu_2O$) species, in relation to ligand and strain effects. However, the difference in chemical properties and electronic orbitals between Pr and Cu caused the differences in the catalytic effects of $Ni_9Mo_1(Pr_2O_3)_{0.375}$ and $Ni_4Mo@Cu_2O$. In order to address this issue, further theoretical and experimental research is required to obtain an atomic and molecular-level understanding of the physical and chemical processes involved in the catalytic decomposition of $N_2H_4·H_2O$ [32].

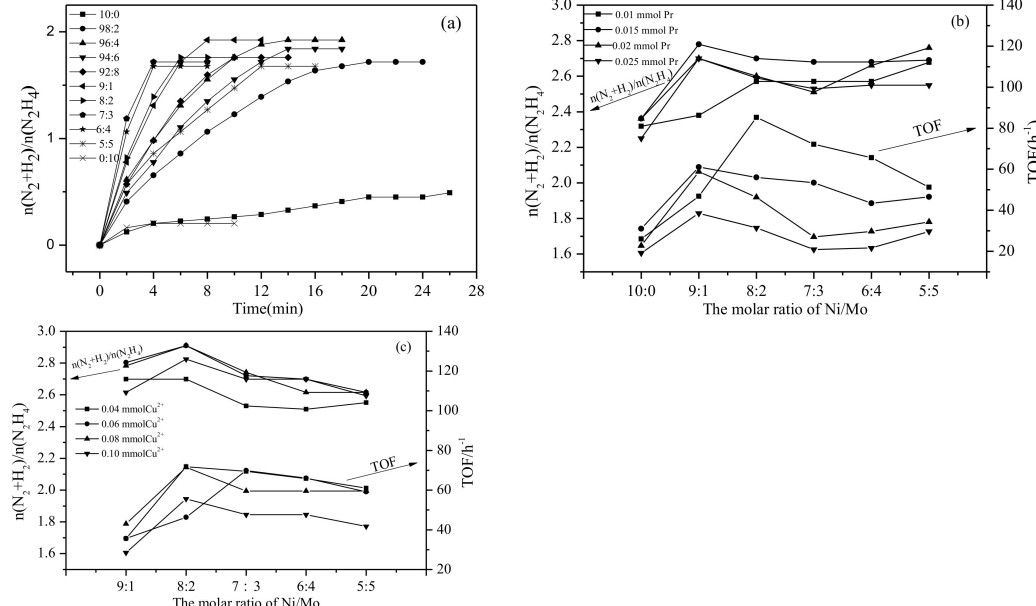

**Figure 4.** (**a**) Time-course profiles for the $N_2H_4·H_2O$ decomposition over the $Ni_{1-x}Mo_x$ catalysts; (**b**) catalytic performance tests of $NiMo(Pr_2O_3)$ with different molar ratios of NiMo and Pr; and (**c**) catalytic performance tests of $NiMo@Cu_2O$ with different molar ratios of NiMo and Cu. All the experiments above were done in the presence of 0.2 mL of 10 M $N_2H_4·H_2O$ and 6 mM NaOH at 343 k.

**Table 1.** Catalytic activity of different catalysts in hydrazine dehydrogenation.

| Catalyst | Temperature(K) | TOF($h^{-1}$) | $E_a$ (kJ $mol^{-1}$) | Ref. |
|---|---|---|---|---|
| $Rh_{4.4}Ni$/graphene | 298 | 28 | - | [12] |
| $Ni_{0.9}Pt_{0.1}/Ce_2O_3$ | 298 | 28.1 | 42.3 | [33] |
| $Ni_{1.5}Fe_{1.0}/(MgO)_{3.5}$ | 299 | 11 | - | [34] |
| $NiIr_{0.059}/Al_2O_3$-HT | 303 | 12.4 | 49.3 | [17] |
| $NiPt_{0.057}/Al_2O_3$-HT | 303 | 16.5 | 34 | [16] |
| $Ni_{0.6}Fe_{0.4}Mo$ | 323 | 28.8 | 50.7 | [20] |
| $Cu@Fe_5Ni_5$ | 343 | 11.9 | - | [27] |
| $NiFe/Cu$ | 343 | 17.6 | 45.95 | [28] |
| $Ni_9Mo_1(Pr_2O_3)_{0.375}$ | 343 | 62 | 43.24 | This work |
| $Ni_4Mo@Cu_2O$ | 343 | 71.4 | 46.47 | This work |

Alkali concentration plays a critical role in promoting the selective decomposition of $N_2H_4·H_2O$ to generate $H_2$. For $N_2H_4·H_2O$-based hydrogen generation using $Ni_9Mo_1(Pr_2O_3)_{0.375}$ or $Ni_4Mo@Cu_2O$ catalysts, the optimal alkali concentration was determined to be 6 mM. As shown in Figure 5a, when the concentration of NaOH in the system was 6 mM, the TOF achieved over the $Ni_9Mo_1(Pr_2O_3)_{0.375}$ catalyst was almost ten-fold higher than that obtained in the absence of NaOH, whereas an increase of H2 selectivity from 25% to 92% was also observed. $H_2$ selectivity from 25% to 92%, compared with

catalytic system with no added NaOH. Similarly, it can be seen from Figure 5b that the $Ni_4Mo@Cu_2O$ catalyst also showed better performance when 6 mM NaOH was added, where the $H_2$ selectivity increased to 97% and the TOF increased from 4 $h^{-1}$ to 71.4 $h^{-1}$. The hydrogen production rate and $H_2$ selectivity show no further change with further increases in the Na OH concentration. We surmise that the promoting effects of alkali on both catalytic activity and selectivity might be associated with the enhancement of the reducibility of $N_2H_4$. As seen in Equations (1) and (2), increasing $OH^-$ concentration not only enhances conversion from $N_2H_5^+$ to $N_2H_4$, but is also responsible for decreasing the basic byproduct $NH_3$, enhancing the selectivity of $H_2$ [22,32,35]. Furthermore, high concentration of $OH^-$ enhanced the hydrolysis of $N_2H_4 \cdot H_2O$, while the decomposition rate of catalytic substrate is improved.

$$NH_3 + H_2O \leftrightarrow NH_4^+ + OH^- \tag{1}$$

$$N_2H_5^+ + OH^- \leftrightarrow N_2H_4 + H_2O \leftrightarrow N_2 + H_2 + H_2O \tag{2}$$

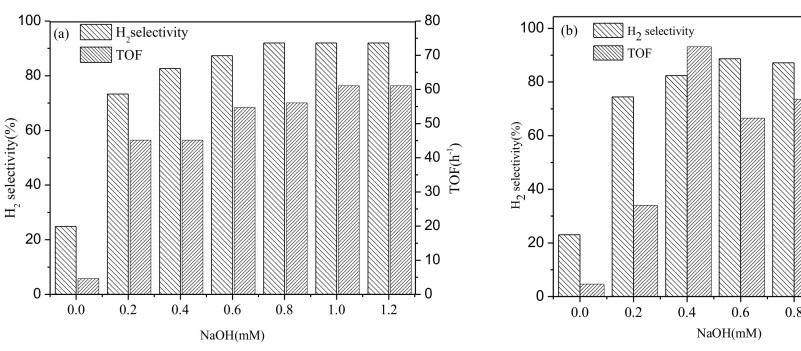

**Figure 5.** Influence of NaOH concentration on $H_2$ selectivity and reaction rate of the catalysts (**a**) $Ni_9Mo_1(Pr_2O_3)_{0.375}$ and (**b**) $Ni_4Mo@Cu_2O$.

The $N_2H_4 \cdot H_2O$ gas production curve and the Arrhenius diagram of ln(TOF) and 1/T from 298 to 343 K are shown in Figure 6. It can be seen from Figure 6a,c that the catalysts show poor catalytic activity at room temperature, and the gas production rate and hydrogen production of $N_2H_4 \cdot H_2O$ increase gradually with increasing reaction temperature. When the reaction temperature is 80 °C, the reaction rate and gas production are not changed, and so a reaction temperature of 70 °C is used in the experiments described in this paper. The Arrhenius plot of ln(TOF) vs. 1/T for our catalysts are plotted in Figure 6b,d, from which the calculated activation energy (Ea) values were 43.24 kJ $mol^{-1}$ and 46.47 kJ $mol^{-1}$, for $Ni_9Mo_1(Pr_2O_3)_{0.375}$ and $Ni_4Mo@Cu_2O$, respectively. This value compares favorably with the literature results for $Ni_{0.99}Pt_{0.01}$ (49.95 kJ $mol^{-1}$) [14], $Ni-Al_2O_3-HT$ (49.3 kJ $mol^{-1}$) [36], $Ni_{88}Pt_{12}/MIL-101$(55.5 kJ $mol^{-1}$) [10], and the $Ni_{0.9}Cr_{0.1}$ (63.5 kJ $mol^{-1}$) and $Ni_{0.8}W_{0.2}$ (47.3 kJ $mol^{-1}$) catalysts [19], indicating that the $Ni_9Mo_1(Pr_2O_3)_{0.375}$ and $Ni_4Mo@Cu_2O$ catalysts have a superior catalytic performance in terms of degrading $N_2H_4 \cdot H_2O$.

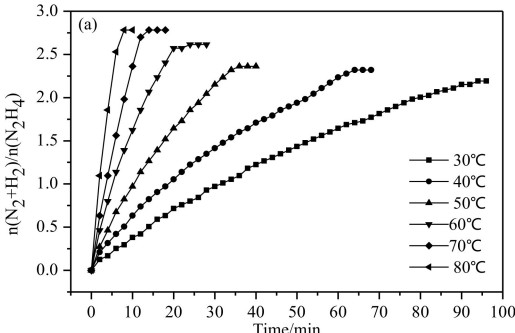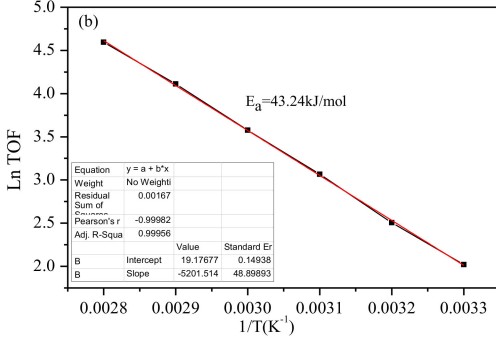

**Figure 6.** *Cont.*

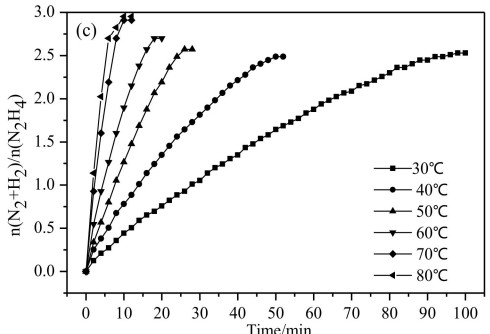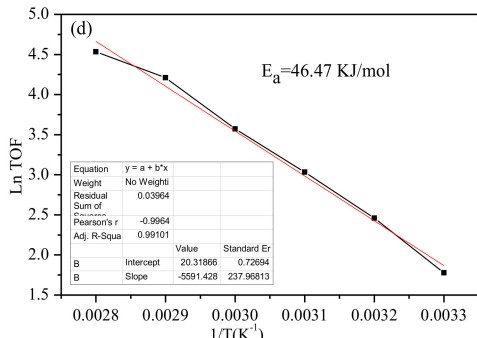

**Figure 6.** Influence of reaction temperature on the $N_2H_4 \cdot H_2O$ decomposition at the examined temperature range from 30 °C to 80 °C and Arrhenius treatment of the temperature-dependent rate data for determination of activation energy of the catalysts (**a,b**) $Ni_9Mo_1(Pr_2O_3)_{0.375}$ and (**c,d**) $Ni_4Mo@Cu_2O$.

High durability and stability are crucial to the practical application of a catalyst. Therefore, in the present study, durability tests were performed on $Ni_9Mo_1(Pr_2O_3)_{0.375}$ and $Ni_4Mo@Cu_2O$ for the same decomposition reaction at 343 K, by adding aqueous hydrous hydrazine (2 mM) to the catalyst after the last reaction round is completed. Time-course plots for the decomposition of $N_2H_4 \cdot H_2O$ that was catalyzed by $Ni_9Mo_1(Pr_2O_3)_{0.375}$ are shown in Figure 7a, while Figure 7b shows the corresponding plots for $Ni_4Mo@Cu_2O$. As shown in Figure 7, the catalytic activity of the $Ni_9Mo_1(Pr_2O_3)_{0.375}$ catalyst exhibits no significant decline even after four rounds of decomposition reactions, indicating that the catalyst is highly stable. However, the $Ni_4Mo@Cu_2O$ catalyst exhibits obvious activity attenuation throughout cyclic usage. We may speculate that this observed activity decay could be attributed to the following possible mechanisms. First, the structure and morphology of the catalysts may have changed because as the number of cycles increases, both the partial agglomeration of metal nanoparticles and amount of exposed active sites decrease. Second, the decay could be caused by the surface oxidation of the catalysts, although as the catalyst is not cleaned by centrifugal separation during the catalytic process, which reduces the area and time of contact of the catalyst with air, this is less likely than the first potential mechanism [37,38].

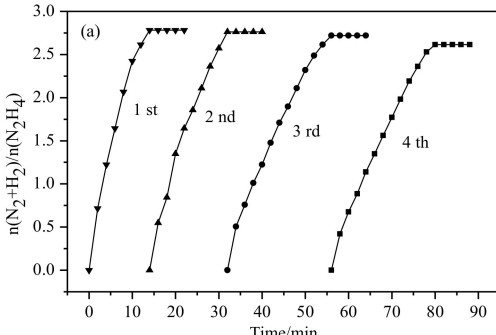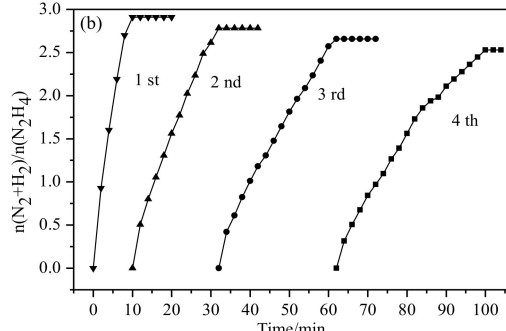

**Figure 7.** (**a**) Durability tests of $Ni_9Mo_1(Pr_2O_3)_{0.375}$ on decomposition of hydrazine at 343 K and (**b**) durability tests of $Ni_4Mo@Cu_2O$ on decomposition of hydrazine at 343 K.

## 3. Materials and Methods

### 3.1. Reagents and Materials

All chemicals were obtained from commercial sources and used without further purification. Nickel (II) chloride hexahydrate ($NiCl_2 \cdot 6H_2O$, Aladdin reagent Co., Ltd., Shanghai, China, >98%), sodium molybdate dehydrate ($Na_2MoO_4 \cdot 2H_2O$, Tianjin Kemiou Chemical Reagent Co., Ltd., Tianjin, China, 99.5%), praseodymium (III) nitrate hexahydrate ($PrN_3O_9 \cdot 6H_2O$, Aladdin reagent Co., Ltd., 99%),

copper sulfate pentahydrate (CuSO$_4$ 5H$_2$O, Tianjin Kemiou Chemical Reagent Co., Ltd., Tianjin, China, 99%), hydrous hydrazine (N$_2$H$_4$·H$_2$O, Aladdin reagent Co., Ltd., Shanghai, China, 50%), sodium borohydride (NaBH$_4$, Tianjin Fengchuan Chemical Reagent Co., Ltd., Tianjin, China, 99%), and sodium hydroxide (NaOH, Tianjin Kemiou Chemical Reagent Co., Ltd., Tianjin, China, 99%). Ultrapure water was obtained by reversed osmosis followed by ion exchange and filtration.

### 3.2. Characterization

Powder XRD data were obtained using an X-ray diffractometer with Cu k$\alpha$ radiation, at an operational voltage and current maintained at 40 kV and 40 mA, respectively. The scanning range for the spectra was 2$\theta$ = 10–90°, at a scanning rate of 8°/min. The sample morphologies and sizes were analyzed using a JEM-2001F field emission transmission electron microscope, manufactured by JEOL Ltd., Tokyo, Japan. XPS measurements were conducted on an ESCALABMKLL 250Xi spectrometer using an aluminum k$\alpha$ source. The energy of the radiation was h$\nu$ = 1486.6 eV, while the spectrometer was operated at 12.5 kV and 16 mA. The Ar sputtering experiments were performed under a background vacuum of $8 \times 10^{-10}$ Pa. The binding energies were calibrated using the C 1 s peak (284.6 eV) of adventitious carbon, and curve fitting was performed using the XPS PEAK 4.1 software. Particle size distribution was measured using a dynamic light scattering nanometer laser particle size analyzer (GS90). The catalyst powder was evenly dispersed in the ultra-pure water by ultrasonic waves before the particle size distribution of the solution was analyzed.

### 3.3. Synthesis of Ni-Based Catalysts: Ni$_{1-x}$Mo$_x$(Pr$_2$O$_3$)$_y$ and Ni$_{1-x}$Mo$_x$@(Cu$_2$O)$_y$

Ni-based nanocatalysts (Ni$_{1-x}$Mo$_x$(Pr$_2$O$_3$)$_y$ and Ni$_{1-x}$Mo$_x$@(Cu$_2$O)$_y$) were prepared by a coreduction method. The typical synthesis procedures for Ni$_9$Mo$_1$ (Pr$_2$O$_3$)$_{0.375}$ and Ni$_4$Mo@Cu$_2$O are described here; 1.8 mL of NiCl$_2$·6H$_2$O (0.1 mol L$^{-1}$) and 0.2 mL of Na$_2$MoO$_4$·2H$_2$O (0.1 mol L$^{-1}$) were added to a round-bottomed flask mixing uniformity, and 0.15 mL of PrN$_3$O$_9$·6H$_2$O (0.1 mol L$^{-1}$) was then added to the above mixture solution. 1.5 mL of NaBH$_4$ (0.03 g mL$^{-1}$) was then quickly added to the above mixture solution for reduction. The reaction solution was vigorously stirred until bubble generation ceased, resulting in the generation of a black suspension of the Ni$_9$Mo$_1$(Pr$_2$O$_3$)$_{0.375}$ catalyst, which was directly used in catalytic reactions. For the Ni$_4$Mo@Cu$_2$O catalyst, the same preparation procedure as for Ni$_9$Mo$_1$(Pr$_2$O$_3$)$_{0.375}$ was adopted, where 0.8 mL (0.1 mol l$^{-1}$) of CuSO$_4$ 5H$_2$O was used instead of the PrN$_3$O$_9$·6H$_2$O. It is of note that Cu must be added before the Na$_2$MoO$_4$·2H$_2$O, where the Na$_2$MoO$_4$·2H$_2$O was alkaline, Cu$^{2+}$ was easily precipitated when it encountered an alkaline solution, and so after adding the Na$_2$MoO$_4$·2H$_2$O, NaBH$_4$ was quickly added to avoid precipitation. Ni$_{1-x}$Mo$_x$(Pr$_2$O$_3$)$_y$ and Ni$_{1-x}$Mo$_x$@(Cu$_2$O)$_y$ catalysts with other molar ratios were prepared using the same procedure by adjusting the relative amounts of precursors, and these catalysts were used for characterization tests after vacuum drying.

### 3.4. Catalysis of Hydrazine Hydrate to Produce Hydrogen

The catalytic reaction of hydrous hydrazine to produce hydrogen by Ni$_{1-x}$Mo$_x$(Pr$_2$O$_3$)$_y$ and Ni$_{1-x}$Mo$_x$@(Cu$_2$O)$_y$ were tested at 343 K with vigorous stirring. The catalysts were kept in a 50-mL two-neck round-bottomed flask, one neck was connected to a gas burette to monitor the volume of H$_2$ released from hydrous hydrazine hydrolysis, while the other neck was used for the introduction of hydrous hydrazine (0.2 mL, 10 mol L$^{-1}$). In order to maintain the reaction temperature, the flask was placed in a thermostat equipped with a water circulating system, and the reaction was determined to be complete when there was no further gas generation. The molar ratios of n(metal)/n(N$_2$H$_4$) were theoretically set to 0.11 and 0.14 for the catalytic reactions of Ni$_9$Mo$_1$(Pr$_2$O$_3$)$_{0.375}$ and Ni$_4$Mo@Cu$_2$O,

respectively. The selectivity towards $H_2$ generation (X) can be calculated using Equation (3), while the TOF was calculated using Equation (4).

$$X = \frac{3\lambda - 1}{8} \left[ \lambda = \frac{n[H_2 + N_2]}{n[N_2H_4]} \right] \tag{3}$$

$$TOF = \frac{n(N_2H_4)_{50\%}}{n(NPs)*t_{50\%}} \tag{4}$$

where TOF refers to the initial turnover frequency, $n(N_2H_4)_{50\%}$ is the amount of hydrous hydrazine when it is half dissolved, n (nanocatalysts) is the mole amount of the metal, and $t_{50\%}$ is the reaction time when conversion reached 50%.

### 3.5. Durability Testing for the Catalysts

In order to test the durability of the $Ni_9Mo_1(Pr_2O_3)_{0.375}$ and $Ni_4Mo@Cu_2O$ catalysts, after the hydrogen generation reaction was completed the first time, 0.2 mL of hydrazine monohydrate was subsequently added into the reaction flask. Such cycles for testing the $N_2H_4 \cdot H_2O$ decomposition of the catalyst were conducted four times at 343 K.

## 4. Conclusions

In summary, we demonstrated a facile coreduction approach to preparing $Pr_2O_3$ and $Cu_2O$-doped NiMo catalysts, without the need for any surfactant or support material, under ambient conditions. The resulting $Ni_9Mo_1(Pr_2O_3)_{0.375}$ and $Ni_4Mo@Cu_2O$ catalysts exhibit good catalytic performances in terms of $N_2H_4 \cdot H_2O$ decomposition, which surpassed that of NiMo catalysts that were produced without $Pr_2O_3$ or $Cu_2O$ doping. The effects of material ratios, NaOH concentration, and reaction temperature on the catalytic efficiency of the catalysts were studied through characterization and catalytic performance evaluation, and compared with unmodified NiMo catalysts. The results show that in the process of modifying NiMo, the $NiMoPr_2O_3$ is amorphous in structure, and characterized by low crystallinity and an increased number of oxygen vacancies that are beneficial to active site exposure and particle dispersion. $NiMo@Cu_2O$ exhibited a core–shell structure, in which $Cu_2O$ is the shell and NiMo is the core, and the electrons in the core and shell are coupled to each other, with an electron-transfer efficiency that improves catalytic efficiency. Among all the prepared catalysts, $Ni_9Mo_1(Pr_2O_3)_{0.375}$ and $Ni_4Mo@Cu_2O$ showed the highest $H_2$ selectivity in the presence of 0.2 mL of 10 M $N_2H_4 \cdot H_2O$ and 6 mM NaOH at 343 K, with TOFs of 62 $h^{-1}$ and 71.4 $h^{-1}$, respectively. These values are higher than the majority of reported results for similar systems The diameters of these well-dispersed ultrafine catalysts were approximately 5.5–5.9 nm and 6.0–6.3 nm, respectively, lower than that of NiMo catalyst particles (approximately 6.7 nm), which led to an increase in the contact area between $N_2H_4 \cdot H_2O$ and the catalyst, and promoted decomposition. This remarkable improvement in the catalytic performance of precious metal-free nanocatalysts demonstrates a promising strategy for future hydrogen production from hydrous hydrazine.

**Author Contributions:** Conceptualization, Y.L. and N.S.; Methodology, H.Z.; Validation, C.M.; Formal Analysis, H.Z.; Investigation, H.Z.; Writing—Original Draft Preparation, Y.L. and H.Z.; Writing—Review & Editing, Y.L., C.M., and N.S.; Supervision, Y.L. and C.M.; Project Administration, Y.L. and N.S.; Funding Acquisition, Y.L. and C.M.

**Funding:** This work was kindly supported by the National Natural Science Foundation of China (51508384 and 51508383), the Natural Science Foundation of Tianjin of China (18JCQNJC09000), the Science and Technology Plans of Tianjin of China (17YFCZZC00360), and the Research Fund of the Tianjin Key Laboratory of Aquatic Science and Technology (No. TJKLAST-ZD-2017-03).

**Conflicts of Interest:** The authors declare no conflicts of interest.

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
