# Peer review of "Modified Nimo Nanoparticles for Efficient Catalytic Hydrogen Generation from Hydrous Hydrazine"

_catalysts, doi:10.3390/catal9070596_

Round 1

Reviewer 1 Report

“Modified NiMo nanoparticles for efficient catalytic hydrogen generation from Hydrous Hydrazine” depicts an interesting manuscript within the scope of Catalysts. The authors demonstrate the synthesis and characterization of two different catalyst systems (Ni9Mo1(Pr2O3)0.375 and Ni4Mo@Cu2O) for hydrogen generation from hydrazine. The catalysts are characterized by XRD, XPS, TEM and catalytic properties measurements.

While I like the general outline and the concept of the manuscript, I have a several issues which should be addressed:

General:

Typographical and grammatical errors need to be corrected and the language should be improved significantly.

Introduction:

Literature is discussed thoroughly and the topic is introduced nicely.

Results and Discussion:

XRD: I agree on the possible amorphous state of the Pr2O3 phase which might affect the width and the intensity of the reflection corresponding to the Ni phase. However, have you measured the FWHM of the Ni reflection and the exact position of its maximum? This might help to identify the influence of an amorphous phase on the NiMo nanoparticles.

On the other hand, why is the Cu2O reflection intense and sharp? The FWHM of the Cu2O reflection actually indicates larger particles (Derived from Scherrer Equation which connects the defects and thus the FWHM with the nanoparticle size). Can you explain this discrepancy and why no larger particles can be observed in the TEM images? However, I disagree on the statement, that no Ni reflection is visible for the Ni4Mo@Cu2O nanoparticles. The Cu2O reflections might overlay and thus conceal the Ni reflections.

TEM: How do you obtain your particle size distributions and the standard deviations? How many nanoparticles are counted? The size distributions look artificial. How do you actually define a nanoparticle (for me it is not obvious to discriminate between several nanoparticles in your TEM images? You have to explain your procedure in the experimental section. How do you explain the size discrepancy between Ni9Mo1 and Ni9Mo1(Pr2O3)0.375 nanoparticles (while still maintaining standard deviations of 0.01 nm)?

XPS: There are multiple errors here: You describe a correction of the spectra with adventitious carbon to 284.6 eV. However, there is a shift between the Ni4Mo@Cu2O) nanoparticles and the Ni9Mo1 and Ni9Mo1(Pr2O3)0.375 nanoparticles of aroun 1 eV in Ni as well as Mo spectra. Such a large chemical shift does not seem logical to be caused by Cu2O. You should reavaluate and preferably remeasure your spectra.

You should indicate the oxidation state of your Ni.

The fits are not deconvolute the spectra properly. Please take your time and redo the fits.

The Mo spectra are not evaluated correctly. The Mo 3d5/2 peak is always labelled Mo(VI) and the Mo 3d3/2 peak is always labelled Mo(V). Both peaks should correspond to Mo(VI) but are just different orbitals. The oxygen in Mo(V) should actually less strongly chemically shift the binding energy of 3d5/2 electrons than the oxygen in Mo(VI).

For Pr, only the 3d5/2 is shown. Here, the spectra should be extended to the region of Pr 3d3/2.

Materials and Methods:

XRD: no spectra but diffractograms.

Describe your TEM evaluation/Counting of particles.

Author Response

General:

Q: Typographical and grammatical errors need to be corrected and the language should be improved significantly.

Reply: Thanks for your suggestion. Typographical and grammatical errors has been corrected now and the language has also already been improved significantly, which has been marked as “red” in the revised paper

Introduction: Literature is discussed thoroughly and the topic is introduced nicely.

Q: Results and Discussion

Q: XRD: I agree on the possible amorphous state of the Pr2O3 phase which might affect the width and the intensity of the reflection corresponding to the Ni phase. However, have you measured the FWHM of the Ni reflection and the exact position of its maximum? This might help to identify the influence of an amorphous phase on the NiMo nanoparticles.

On the other hand, why is the Cu2O reflection intense and sharp? The FWHM of the Cu2O reflection actually indicates larger particles (Derived from Scherrer Equation which connects the defects and thus the FWHM with the nanoparticle size). Can you explain this discrepancy and why no larger particles can be observed in the TEM images? However, I disagree on the statement, that no Ni reflection is visible for the Ni4Mo@Cu2O nanoparticles. The Cu2O reflections might overlay and thus conceal the Ni reflections.

Reply: Thanks.

1) According to the standard spectrum of the crystalline Ni (PDF # 04-0850), Ni has diffraction peaks with semi-peak width of 0.2458, 0.2838, and 0.4013 at 2θ = 44.5 °, 51.8 °, and 76.4 °, respectively. The prepared NiMo catalyst in this study, only showed a weaker diffraction peak at about 2θ = 44.5 °, and it only corresponds to one diffraction peak in the crystalline Ni standard spectrum. So the peak and semi-peak width could not be accurately determined through the Jade software.

2) According to Jade software analysis, the diffraction peaks of the prepared Ni4Mo@Cu2O catalyst could completely coincide with that of the crystal state Cu2O at 2θ = 36.4 °, 41.3 °, 61.3 °, and 73.5 °. It shows that there were obvious Cu2O crystals in the prepared Ni4Mo@Cu2O catalyst. The diffraction peak of Cu2O was strong and sharp, indicating that the structure of the Cu2O crystal was complete. However, there was some adhesion and overlap between the prepared Ni4Mo@Cu2O catalyst particles, then it was impossible to see larger and clearer particles during low-power TEM tests.

As for the explanation that the Cu2O diffraction summit was superimposed to cover up the Ni diffraction peak, I personally thought the possibility was little. Because the diffraction peak position of the Cu2O diffraction peak was different from that of the Ni diffraction peak, the possibility of interference between them should also be relatively little.

Q: TEM: How do you obtain your particle size distributions and the standard deviations? How many nanoparticles are counted? The size distributions look artificial. How do you actually define a nanoparticle (for me it is not obvious to discriminate between several nanoparticles in your TEM images? You have to explain your procedure in the experimental section. How do you explain the size discrepancy between Ni9Mo1 and Ni9Mo1(Pr2O3)0.375 nanoparticles (while still maintaining standard deviations of 0.01 nm)?

Reply: Thanks.

1) The particle size was determined by a dynamic light scattering particle size distribution tester. The catalyst powder was dispersed into the ultrapure water by ultrasonic and the uniform solution was tested then. The determination of the standard deviation was a mistake due to our own judgment and we will certainly do avoid such mistakes in the future. Therefore, we have removed the artificial distribution fitting in the distribution of particle size in figures 2g, 2h, and 2i. The figure shows only the particle size distribution measured by the tester.

2) Through XRD and TEM analysis, the amorphous degree of the Ni9Mo1(Pr2O3)0.375 catalyst increased, the dispersion was more uniform. Combined with the results of the particle size distribution, it was indicated that there was a difference between the two catalysts.

Q: XPS: There are multiple errors here: You describe a correction of the spectra with adventitious carbon to 284.6 eV. However, there is a shift between the Ni4Mo@Cu2O) nanoparticles and the Ni9Mo1 and Ni9Mo1(Pr2O3)0.375 nanoparticles of aroun 1 eV in Ni as well as Mo spectra. Such a large chemical shift does not seem logical to be caused by Cu2O. You should reavaluate and preferably remeasure your spectra.

Reply: Thanks for this import suggestion. It was indeed our fault and then we have reavaluated and corrected the spectra of the Ni9Mo1, Ni9Mo1(Pr2O3)0.375 and Ni4Mo@Cu2O catalysts and have now replaced figures 3a, 3d and 3e with corrected images.

Q: You should indicate the oxidation state of your Ni.

Reply: Thanks. The oxidation state of your Ni has been supplied in Fig. 3a.

Q: The fits are not deconvolute the spectra properly. Please take your time and redo the fits.

Reply: Thanks for the important suggestion. It was indeed our fault. Therefore, we have re-separated the peak and fitted the XPS diagram of each element in the catalysts, and then have replaced figures 3a, 3d and 3e with corrected images.

Q: The Mo spectra are not evaluated correctly. The Mo 3d5/2 peak is always labelled Mo(VI) and the Mo 3d3/2 peak is always labelled Mo(V). Both peaks should correspond to Mo(VI) but are just different orbitals. The oxygen in Mo(V) should actually less strongly chemically shift the binding energy of 3d5/2 electrons than the oxygen in Mo(VI).

Reply: Thanks. We have replaced Mo(V) by Mo(VI) in Fig. 3b, Fig. 3c, Fig. 3d according to the suggestion.

Q: For Pr, only the 3d5/2 is shown. Here, the spectra should be extended to the region of Pr 3d3/2.

Reply: Thanks. According to the information we found, Pr only displays the 3d5/2 track, without 3d3/2 track. Therefore, the spectra was not extended to the region of Pr 3d3/2. The following Fig. is the supplemental information.

Q: Materials and Methods:

Q: XRD: no spectra but diffractograms.

Reply: We did not understand this comment. There were indeed diffractograms without spectra in this research.

Q: Describe your TEM evaluation/Counting of particles.

Reply: The crystal d space was calculated using Digital Micrograph's software. The detailed operation was as follows: Input the TEM image, select the range that needs to be analyzed, click the dotted box in the ROI toys, hold down the ALT key, select a square of the right size around the analyzed area. Then choose the FFT under the Process menu to obtain the diffraction pattern of the reciprocal space. The dotted line in the ROI toys is selected, a straight line symmetrical to the center spot is drawn, and the two ends accurately fall to the two symmetrical diffraction points (the two points of the nearest central spot), and then the Control dialogue box. Get the value of L, divided by 2, take the reciprocal is the d space value.

Reviewer 2 Report

The manuscript (catalysts-506108) entitled “Modified NiMo nanoparticles for efficient catalytic hydrogen generation from Hydrous Hydrazine” seems to describe an interesting work. However, the quality of English used is very low (a lot of syntax and grammatical errors). Thus, it is difficultly understandable. For this reason, I cannot propose publication in the present form.  

Author Response

Thanks for  comments . We have polished the whole paper to make it more suitalbe to the journal.

Reviewer 3 Report

Reviewer’s Report

Manuscript ID: catalysts-506108

Title: Modified NiMo nanoparticles for efficient catalytic hydrogen generation from Hydrous Hydrazine

In this work, the authors have reported the precious-metal-free NiMoM (M= Pr2O3, Cu2O) catalysts for the catalytic hydrogen generation from Hydrous Hydrazine. The results present in this manuscript are interesting and I would recommend the manuscript for publication after the following revisions.

1)      Calculate the crystallite size of the samples using XRD.

2)      Page 3, lines 126-129; “The 5d empty orbit of Pr is a space station for electron transfer the synthesis catalyst will disturb Ni electron configuration, make the crystal shape change. While Ni, Cu, and Mo are transition metals, the difference of oxidation ability plays an important role in the formation of the structure”. Authors should provide a suitable reference for these statements.

3)      Provide the EDS color mappings to know the distribution of elements in the samples.

4)      Page 5, lines 159-161; “From Fig. 3.e, one prominent peak at 932.9 eV is readily assigned to the Pr ( ) 3d5/2, indicating that NaBH4 did not reduce Pr3+ to Pr in Ni9Mo1(Pr2O3)0.375 during the catalyst preparation,” Authors should use the following references to support this statement.

RSC Adv. 2015,5, 30275-30285; RSC Adv. 2016, 6, 44826–44837

5)      Provide the characterizations for the spent catalysts in order to understand the structural stability of the samples after the reaction.

6)      There are some typo/grammatical errors that should be minimized before publication.

Author Response

Q: Calculate the crystallite size of the samples using XRD.

Reply: Because the prepared NiMo and Ni9Mo1(Pr2O3)0.375 was amorphous (relatively low crystallization), it was impossible to locate the semi-peak width of the exact peak position using Jade software to seek. Therefore, the particle size could not be accurately calculated using XRD. Ni4Mo@Cu2O catalyst was a core-shell structure, and the complete crystal structure of shell could be calculated the particle size. However, the core was amorphous and dispersed, it was either impossible to obtain the accurate particle size. We tried to find the characteristic peaks using Jade software, but they could not be accurately located due to the deviation each time. We calculated that the particle size reached about 100 nanometers through the Scherrer formula. We concern the value was inaccurate.

Q: Page 3, lines 126-129; “The 5d empty orbit of Pr is a space station for electron transfer the synthesis catalyst will disturb Ni electron configuration, make the crystal shape change. While Ni, Cu, and Mo are transition metals, the difference of oxidation ability plays an important role in the formation of the structure”. Authors should provide a suitable reference for these statements. 

Reply: Thanks. We has provided the suitable references for these statements as follows:

Men, Y.; Du, X.; Cheng, G.; Luo, W. CeOx-modified NiFe nanodendrits grown on rGO for efficient catalytic hydrogen generation from alkaline solution of hydrazine. International Journal of Hydrogen Energy 2017, 42, 27165-27173, doi:10.1016/j.ijhydene.2017.08.214.

Manukyan, K.V.; Cross, A.; Rouvimov, S.; Miller, J.; Mukasyan, A.S.; Wolf, E.E. Low temperature decomposition of hydrous hydrazine over FeNi/Cu nanoparticles. Applied Catalysis A: General 2014, 476, 47-53, doi:10.1016/j.apcata.2014.02.012.

Yao, Q.; Lu, Z.H.; Zhang, R.; Zhang, S.; Chen, X.; Jiang, H.L. A noble-metal-free nanocatalyst for highly efficient and complete hydrogen evolution from N2H4 BH3. Journal of Materials Chemistry A 2018, 6, 4386-4393, doi:10.1039/c7ta10886a.

Q: Provide the EDS color mappings to know the distribution of elements in the samples. 

Reply: In the process of the revision of the paper, we replenished to make the EDS imagines of the NiMo@Cu2O catalyst (as follows). However, the oxygen element was not analyzed during the EDS test of the NiMoPr catalyst and the correct EDS diagram of the NiMoPr catalyst was not obtained in time. Due to limited time, it is too late to make up this EDS analysis test in this paper. We feel sorry about this paper and we must take care of avoiding such mistakes in our future research.

 (NiMoPr catalyst)

(Core of NiMo@Cu2O catalyst)

(Shell of NiMo@Cu2O catalyst)

Q: Page 5, lines 159-161; “From Fig. 3.e, one prominent peak at 932.9 eV is readily assigned to the Pr ( Ⅲ) 3d5/2, indicating that NaBH4 did not reduce Pr3+ to Pr in Ni9Mo1(Pr2O3)0.375 during the catalyst preparation,” Authors should use the following references to support this statement.

Reply: Thanks. We have supplied the following references to support above statements in the revised manuscrpt.

Devaiah, D.; Thrimurthulu, G.; Smirniotis, P.G.; Reddy, B.M. Nanocrystalline alumina-supported ceria-praseodymia solid solutions: Structural characteristics and catalytic CO oxidation. RSC Advances 2016, 6, 44826-44837, doi:10.1039/c6ra06679h.

Devaiah, D.; Tsuzuki, T.; Boningari, T.; Smirniotis, P.G.; Reddy, B.M. Ce0.80M0.12Sn0.08O2-δ(M = Hf, Zr, Pr, and La) ternary oxide solid solutions with superior properties for CO oxidation. RSC Advances 2015, 5, 30275-30285, doi:10.1039/c5ra00557d.

Q: Provide the characterizations for the spent catalysts in order to understand the structural stability of the samples after the reaction.

Reply: Thanks for the useful suggestion. Considering the characterizations for the spent catalysts, it is necessary to recycle the same catalysts with the same preparation conditions and reaction conditions many times. The experimental process might be complex and cumbersome, and we indeed did not consider the subsequent problems during the experiments. However, we must pay adequate attention and consideration in the future relevant research.

Q: There are some typo/grammatical errors that should be minimized before publication. 

Reply: The paper has been polished by the Company of Editage as follows:

Reviewer 4 Report

The authors present and single pot synthesis methods for nanostructured catalyst for the decomposition of hydrazine to hydrogen. Significant useful characterization data is generated in addition to data demonstrating the high activity of the materials. However, revisions are needed is several palaces to make the manuscript more clear. In particular it is difficult to fully review the XPS section. Additionally, the authors do not do much to tie the xrd, tem, and XPS results to the relative activities of the materials.

1.      There are several grammatical errors and/or awkward word choices. For example, the second sentence of the abstract start with “And” which is not acceptable in formal writing. Additionally, some sections, such as the top of page 3 and the XPS discussions are particularly hard to follow.  Review someone proficient in English is suggested.

2.      The introduction does contain a reasonable literature review supporting the rational of the current work.  However, given that hydrazine are somewhat controversial as fuels from the stand point of the energy efficacy of hydrazine production, some additional literature support for its feasibility would be beneficial. Given that there are fuel cells that can utilize ammonia, some expansion on why selectivity for hydrogen vs ammonia is beneficial is needed.

3.      The article make several references to Ni1-xMox(Pr2O3)y while at the same time reporting Ni1Mo9(Pr2O3)y. Either the authors intend Ni0.1Mo0.9 or their notation needs to be revised extensively for clarity.

4.      Given the TEM imaging and preparation methodology, it is far from clear that the catalyst made are truly “nanoparticles.” While the TEM and XRD do strongly support that the materials contain nanocrystals, they seem more likely to be nano-composites or multiple nanoparticles embedded in a larger shell. It is possible that the authors intended to merely refer to the NiMo phase. In either case revision for clarity is needed.

5.      “Due to the difference in atomic electronic configuration and redox potential of Pr and Cu, when the NiMo catalyst is modified, Pr is immersed in the lattice of Ni and Cu is coated on the surface of  NiMo to form nanoparticles with different lattice structures.”  This statement needs clear citation of literature for support, expansion for more detailed reasoning by the authors, or removal.

6.      It is surprising that Cu2O is not reduced in the presence of hydrazine. No spent samples appear to have been characterized by XRD or XPS to confirm that metallic copper does not form.  This needs to be addressed. The reducibility of Cu2O, and the potential effect on the material structure could also be responsible for the observe los of activities in some of those catalyst.

7.      The relative activities of the catalyst should be examined more n light of the other characterization data.

Author Response

Q: There are several grammatical errors and/or awkward word choices. For example, the second sentence of the abstract start with “And” which is not acceptable in formal writing. Additionally, some sections, such as the top of page 3 and the XPS discussions are particularly hard to follow.  Review someone proficient in English is suggested.

Reply: The paper has been polished by the Company of Editage as follows:

Q: The introduction does contain a reasonable literature review supporting the rational of the current work.  However, given that hydrazine are somewhat controversial as fuels from the stand point of the energy efficacy of hydrazine production, some additional literature support for its feasibility would be beneficial. Given that there are fuel cells that can utilize ammonia, some expansion on why selectivity for hydrogen vs ammonia is beneficial is needed.

Reply: Thanks. We enhanced this part of elaboration in the revised manuscript (marked as red)

Q: The article make several references to Ni1-xMox(Pr2O3)y while at the same time reporting Ni1Mo9(Pr2O3)y. Either the authors intend Ni0.1Mo0.9 or their notation needs to be revised extensively for clarity.

Reply: Thanks. We enhanced this part of elaboration in the revised manuscript (marked as red).

Q: Given the TEM imaging and preparation methodology, it is far from clear that the catalyst made are truly “nanoparticles.” While the TEM and XRD do strongly support that the materials contain nanocrystals, they seem more likely to be nano-composites or multiple nanoparticles embedded in a larger shell. It is possible that the authors intended to merely refer to the NiMo phase. In either case revision for clarity is needed.

Reply: Thanks. We have replaced all NPs by catalyst/catalysts to make the revised manuscript more accurate according to the suggestion.

Q: “Due to the difference in atomic electronic configuration and redox potential of Pr and Cu, when the NiMo catalyst is modified, Pr is immersed in the lattice of Ni and Cu is coated on the surface of  NiMo to form nanoparticles with different lattice structures.”  This statement needs clear citation of literature for support, expansion for more detailed reasoning by the authors, or removal.

Reply: Thanks for the import suggestion. The useful comments provide the research direction for our future experimental work and we should conduct the detailed study of the relevant content according to your suggestion. We have polished the whole paper and deleted the inaccurate parts.

Q: It is surprising that Cu2O is not reduced in the presence of hydrazine. No spent samples appear to have been characterized by XRD or XPS to confirm that metallic copper does not form.  This needs to be addressed. The reducibility of Cu2O, and the potential effect on the material structure could also be responsible for the observe los of activities in some of those catalyst.

Reply: Thanks for the import suggestion. Because the spent catalyst was not recovered on time, leading to non-characterization of waste samples, it was indeed our fault in the research. Although Hydrazine hydrate is reducible, it is less reducible than NaBH4. In the process of preparing the catalyst, excess reducible NaBH4 is used as a reducing agent, and then slightly Cu2O may be reduced to Cu. It has been demonstrated that the doping of metal Cu with Ni or other non-precious metals can catalyze hydrogen production by liquid hydrogen storage substances [1-3].

1. Li, S.J.; Wang, H.L.; Yan, J.M.; Jiang, Q. Oleylamine-stabilized Cu0.9Ni0.1 nanoparticles as efficient catalyst for ammonia borane dehydrogenation. International Journal of Hydrogen Energy 2017, 42, 25251-25257, doi:10.1016/j.ijhydene.2017.08.120.

2. Zeynizadeh, B.; Mohammadzadeh, I.; Shokri, Z.; Ali Hosseini, S. Synthesis and characterization of NiFe2O@Cu nanoparticles as a magnetically recoverable catalyst for reduction of nitroarenes to arylamines with NaBH4. Journal of Colloid and Interface Science 2017, 500, 285-293, doi:10.1016/j.jcis.2017.03.030.

3. Liang, Z.; Xiao, X.; Yu, X.; Huang, X.; Jiang, Y.; Fan, X.; Chen, L. Non-noble trimetallic Cu-Ni-Co nanoparticles supported on metal-organic frameworks as highly efficient catalysts for hydrolysis of ammonia borane. Journal of Alloys and Compounds 2018, 741, 501-508, doi:10.1016/j.jallcom.2017.12.151.

Q: The relative activities of the catalyst should be examined more n light of the other characterization data.

Reply: Thanks for the import advice. It was indeed our shortcoming in the present research. We must conduct the more detailed study about the relevant content in the future work and research according to your important comments.

Round 2

Reviewer 1 Report

The authors revised their manuscript according to the recommendations. However, there are still a few issues which need to be addressed:

EDX is still mentioned in the experimental section

Abstract:

“The possible formation and growth mechanism of alunite crystallites by ionic liquid-assisted hydrothermal process was proposed and discussed in detail. The reaction conditions, such as presence and the amount of templating argent (ionic liquid), reaction temperature and the composition of the reaction system play considerable roles on the structure and morphologies of the final products.” Which formation and growth model do you propose? This is kept quite vague in the whole discussion part as well. You should add which formation and growth processes you propose for the synthesis of natronalunite in the abstract. Which roles do the temperature and the composition play on structure and morphology?

Answer: Thanks for your kindly comments! On the based our experimental results, we have CrystEngComm Page 2 of 26 proposed simple formation and growth mechanism of the natroalunite nanostructures in the ionic liquid-assisted hydrothermal condition. Of course, this is not complete. According to the reviewers’ comment, we have revised the Abstract section.

There is no hint to the nucleation and growth mechanism in the new revised abstract. However, this is your largest topic within the text.

Introduction

The introduction starts with a general information about formation, growth and the influencing factors for these phases. However, differences in templating agents, interfacial tension and dielectric constants of surrounding media is not discussed any further in the results and discussion part. The whole introduction does not have any central idea why and how this study is conducted. There are only different pieces (General Growth, Natroalunite, Fluoride Adsorption) which are placed together. How do these ideas come together? Why do you need naroalunite for fluoride adsorption, how is the performance of other fluoride adsorbents/methods (maximum load/affinity or efficiency)? Why are you synthesizing natroalunite in this manner and why is the understanding of the growth mechanism important?

While you revised the introduction a little, the introduction still is no unity and no reference systems concerning the adsorption capacity and affinity have been introduced.

Author Response

Dear Reviewer

We are sorry that Comments and Suggestions are not matched with our paper. There is no "the amount of templating argent (ionic liquid)" in the abstract and no "There are only different pieces (General Growth, Natroalunite, Fluoride Adsorption) which are placed together" in the introduction.

There might be something wrong about the "Comments and Suggestions". Therefore, we can not make the point-by-point response to the reviewer’s comments .

Reviewer 2 Report

The revised manuscript has been substantialy improoved. Thus, I can propose publication after minor revision. See specific comments bellow:

1.       Lines 45-46: “Although ammonia can be burned, not only generates less heat than that of H2 as fuel, but also produces harmful gases like NO.” should be written instead of “Although ammonia can be burned, NO as the combustion production, not only generates less heat than that of H2 as fuel, but also produces no unsatisfactory gases.”

2.       Lines 48-51: “Several recent studies found that Ni-based catalysts promoted by noble metals, such as Ni-Rh [7-9], Ni-Pt [10-13], 49 Ni-Pd [14-16], and Ni-Ir, exhibit superior catalytic performances in the decomposition of hydrous hydrazine, with more than 90% H2 selectivity [17,18].” should be written instead of “Several recent studies found that Ni-based catalysts based on noble metals, such as Ni-Rh [7-9], Ni-Pt [10-13], Ni-Pd [14-16], and Ni-Ir, exhibit superior catalytic performances in the decomposition of hydrous hydrazine, with more than 90% H2 selectivity [17,18].”

3.       Lines 157-159: “Doping with Cu leads to a positive shift in the binding energies of the metallic Ni (Ni 2p3/2 852.6 eV), which may be due to a decrease in the electron density and an increase in metal center d-band vacancies [27,28].” should be written instead of “Doping the Cu element leads to a positive shift in the binding energies of the metallic Ni (Ni 2p3/2 852.6 eV), which may be due to a decrease in the electron density and an increase in metal center d-band vacancies [27,28].”

4.       Lines 163-167: “Compared with the 3d5/2 and 3d3/2 peaks that appear at 232.1 and 235.15 eV in NiMo, which are attributed to Mo (VI) species such as MoO3 and MoOx, the corresponding peaks for Ni9Mo1(Pr2O3)0.375 and Ni4Mo@Cu2O are shifted to higher binding energies, indicating that Mo can act as an electron donor for atoms of Ni and Cu.” should be written instead of “Compared with the intense 3d5/2 and 3d3/2 peaks that appear at 232.1 and 235.15 eV in NiMo, which are attributed to Mo (IV) species such as MoO3 and MoOx, the peaks for Ni9Mo1(Pr2O3)0.375 and Ni4Mo@Cu2O are shifted to higher binding energies, indicating that Mo can act as an electron donor for atoms of Ni and Cu.”

5.       Lines 179-180: “Figure 3. X-ray photoelectron spectra of the as-synthesized Ni9Mo1, Ni9Mo1(Pr2O3)0.375, Ni4Mo@Cu2O catalysts: (a) Ni 2p; (b),(c), (d) Mo 3d; (e) Pr 3d; (f) Cu 2p. should be written instead of “Figure 3. spectra of the as-synthesized Ni9Mo1Ni9Mo1(Pr2O3)0.375Ni4Mo@Cu2O: (a) Ni 2p; (b)(c) (d) Mo 3d; (e) Pr 3d; (f) Cu 2p.”

6.       Lines 182-183: “The catalytic performances of the as-synthesized nanocatalysts, in terms of hydrogen generation from N2H4·H2O, are shown in Fig. 4.” should be written instead of “The catalytic performances of the nanocatalysts, in terms of hydrogen generation from N2H4·H2O, compared to that of as-synthesized catalysts are shown in Fig. 4.”

7.       Line 208: “Figure 4. (a) Time-course profiles for the…” should be written instead of “Figure 4. Time-course profiles for the …”

8.       Lines 211, 215, 216, 220 and 342: “mM” should be written instead of “mmol”

9.       Lines 215-218: “As shown in Fig. 5a, when the concentration of NaOH in the system was 6 mM, the TOF achieved over the Ni9Mo1(Pr2O3)0.375 catalyst was almost ten-fold higher than that obtained in the absence of NaOH, whereas an increase of H2 selectivity from 25% to 92% was also observed.” should be written instead of “As shown in Fig. 5a, when the concentration of NaOH in the system was 6 mmol, the conversion rate of Ni9Mo1(Pr2O3)0.375 catalyst to N2H4·H2O exhibited a greater than ten-fold enhancement of percent conversion and increased H2 selectivity from 25% to 92%, compared with catalytic system with no added NaOH.”

10.     Line 231: “Figure 5. Influence of NaOH concentration on H2 selectivity …” should be written instead of “Figure 5. of NaOH concentration on H2 selectivity …”

11.    Figure 5: Please check the x-axis in (a) and (b) in connection with comment 8 above.

12.    Line 242: “kJ” instead of “KJ”

13.    Line 247: “Figure 6. Influence of reaction temperature …” should be written instead of “Figure 6. of reaction temperature …”

14.    Line 266: “Figure 7. (a) Durability tests of Ni9Mo1(Pr2O3)0.375 on decomposition …” should be written instead of “Figure 7. Durability tests of Ni9Mo1(Pr2O3)0.375 on decomposition …”

Author Response

Comments and Suggestions for Authors

The revised manuscript has been substantialy improoved. Thus, I can propose publication after minor revision. See specific comments bellow:

Q1: Lines 45-46: “Although ammonia can be burned, not only generates less heat than that of H2 as fuel, but also produces harmful gases like NO.” should be written instead of “Although ammonia can be burned, NO as the combustion production, not only generates less heat than that of H2 as fuel, but also produces no unsatisfactory gases.”

Reply: Thanks. Then sentence has been re-written.

Q2: Lines 48-51: “Several recent studies found that Ni-based catalysts promoted by noble metals, such as Ni-Rh [7-9], Ni-Pt [10-13], 49 Ni-Pd [14-16], and Ni-Ir, exhibit superior catalytic performances in the decomposition of hydrous hydrazine, with more than 90% H2 selectivity [17,18].” should be written instead of “Several recent studies found that Ni-based catalysts based on noble metals, such as Ni-Rh [7-9], Ni-Pt [10-13], Ni-Pd [14-16], and Ni-Ir, exhibit superior catalytic performances in the decomposition of hydrous hydrazine, with more than 90% H2 selectivity [17,18].”

Reply: Thanks. Then sentence has been re-written.

Q3: Lines 157-159: “Doping with Cu leads to a positive shift in the binding energies of the metallic Ni (Ni 2p3/2 852.6 eV), which may be due to a decrease in the electron density and an increase in metal center d-band vacancies [27,28].” should be written instead of “Doping the Cu element leads to a positive shift in the binding energies of the metallic Ni (Ni 2p3/2 852.6 eV), which may be due to a decrease in the electron density and an increase in metal center d-band vacancies [27,28].”

Reply: Thanks. Then sentence has been re-written.

Q4: Lines 163-167: “Compared with the 3d5/2 and 3d3/2 peaks that appear at 232.1 and 235.15 eV in NiMo, which are attributed to Mo (VI) species such as MoO3 and MoOx, the corresponding peaks for Ni9Mo1(Pr2O3)0.375 and Ni4Mo@Cu2O are shifted to higher binding energies, indicating that Mo can act as an electron donor for atoms of Ni and Cu.” should be written instead of “Compared with the intense 3d5/2 and 3d3/2 peaks that appear at 232.1 and 235.15 eV in NiMo, which are attributed to Mo (IV) species such as MoO3 and MoOx, the peaks for Ni9Mo1(Pr2O3)0.375 and Ni4Mo@Cu2O are shifted to higher binding energies, indicating that Mo can act as an electron donor for atoms of Ni and Cu.”

Reply: Thanks. Then sentence has been re-written.

Q5: Lines 179-180: “Figure 3. X-ray photoelectron spectra of the as-synthesized Ni9Mo1, Ni9Mo1(Pr2O3)0.375, Ni4Mo@Cu2O catalysts: (a) Ni 2p; (b),(c), (d) Mo 3d; (e) Pr 3d; (f) Cu 2p. should be written instead of “Figure 3. spectra of the as-synthesized Ni9Mo1Ni9Mo1(Pr2O3)0.375Ni4Mo@Cu2O: (a) Ni 2p; (b)(c) (d) Mo 3d; (e) Pr 3d; (f) Cu 2p.”

Reply: Thanks. Then sentence has been re-written.

Q6: Lines 182-183: “The catalytic performances of the as-synthesized nanocatalysts, in terms of hydrogen generation from N2H4·H2O, are shown in Fig. 4.” should be written instead of “The catalytic performances of the nanocatalysts, in terms of hydrogen generation from N2H4·H2O, compared to that of as-synthesized catalysts are shown in Fig. 4.”

Reply: Thanks. Then sentence has been re-written.

Q7: Line 208: “Figure 4. (a) Time-course profiles for the…” should be written instead of “Figure 4. Time-course profiles for the …”

Reply: Thanks. Then sentence has been re-written.

Q8: Lines 211, 215, 216, 220 and 342: “mM” should be written instead of “mmol”

Reply: Thanks. All “mmol” has been corrected to “mM” in the revised manuscript.

Q9: Lines 215-218: “As shown in Fig. 5a, when the concentration of NaOH in the system was 6 mM, the TOF achieved over the Ni9Mo1(Pr2O3)0.375 catalyst was almost ten-fold higher than that obtained in the absence of NaOH, whereas an increase of H2 selectivity from 25% to 92% was also observed.” should be written instead of “As shown in Fig. 5a, when the concentration of NaOH in the system was 6 mmol, the conversion rate of Ni9Mo1(Pr2O3)0.375 catalyst to N2H4·H2O exhibited a greater than ten-fold enhancement of percent conversion and increased H2 selectivity from 25% to 92%, compared with catalytic system with no added NaOH.”

Reply: Thanks. Then sentence has been re-written.

Q10: Line 231: “Figure 5. Influence of NaOH concentration on H2 selectivity …” should be written instead of “Figure 5. of NaOH concentration on H2 selectivity …”

Reply: Thanks. Then sentence has been re-written.

Q11: Figure 5: Please check the x-axis in (a) and (b) in connection with comment 8 above.

Reply: Thanks. Fig.5 has been re-drawn.

Q12: Line 242: “kJ” instead of “KJ”

Reply: Thanks. All “KJ” has been changed to “kJ” in the revised manuscript.

Q13: Line 247: “Figure 6. Influence of reaction temperature …” should be written instead of “Figure 6. of reaction temperature …”

Reply: Thanks. Then sentence has been re-written.

Q14:  Line 266: “Figure 7. (a) Durability tests of Ni9Mo1(Pr2O3)0.375 on decomposition …” should be written instead of “Figure 7. Durability tests of Ni9Mo1(Pr2O3)0.375 on decomposition …”

Reply: Thanks. Then sentence has been re-written.

Reviewer 3 Report

Accept in its present form.

Author Response

Thanks for all your comments and help.

Reviewer 4 Report

The authors have addressed the major issues. However, on final revision, I would suggest that the authors consider making more clear that they are evaluating hydrogen generation for combustion, not for fuel cells (or otherwise clarify their use of "combustion"). Some statements tying observed activity and deactivation to characterization data would still greatly improve the quality and impact of the paper.

Author Response

Q: The authors have addressed the major issues. However, on final revision, I would suggest that the authors consider making more clear that they are evaluating hydrogen generation for combustion, not for fuel cells (or otherwise clarify their use of "combustion"). Some statements tying observed activity and deactivation to characterization data would still greatly improve the quality and impact of the paper.

Reply: Thanks for the important suggestion. We have supplied some sentences about hydrogen generation for combustion, not for fuel cells in the introduction of the revised manuscript.

As an environmentally friendly fuel, hydrogen has the advantages of high combustion heat and high combustion speed. In addition to generating water and a little hydrogen azide during the combustion process, it will not produce substances that are harmful to the environment such as carbon oxides and hydrocarbons compounds. Hydrogen has been proposed as crucial to ensuring secure and sustainable energy development [1-3].